# Prevalence and risk factors of hypertension among clients seeking care at Selected Healthcare Facilities in Kenya

Jasmit Shah[1,2]*, Soraiya Manji[1], Cynthia Smith[2], Jamila Nambafu[1,3], Anthony Ochola[1,4], Linda Barasa[1,5], Faraj Amir[6], Husni Abdalla[7], Simeon Jowi[8], Caroline Mithi[9], Rajiv Patel[1], Sayed K. Ali[1]*

1 Department of Medicine, Aga Khan University, Nairobi, Kenya, 2 Brain and Mind Institute, Aga Khan University, Nairobi, Kenya, 3 PCEA Chogoria Hospital, Chogoria, Kenya, 4 Tenwek Hospital, Bomet, Kenya, 5 Avenue Hospital, Nairobi, Kenya, 6 Garissa County Referral Hospital, Garissa, Kenya, 7 Aga Khan University Hospital, Mombasa, Kenya, 8 Kericho County Referral Hospital, Kericho, Kenya, 9 Kenyatta University Teaching and Referral Hospital, Nairobi, Kenya

* sayed.karar@aku.edu (SKA); jasmit.shah@aku.edu (JS)

## Abstract

### Background

Hypertension remains one of the primary risk factors for cardiovascular disease and the leading cause of mortality worldwide. According to the World Health Organization (WHO), as estimated 1.28 billion adults suffer from hypertension worldwide and approximately half are unaware of the problem. This study aimed to explore the prevalence of hypertension, and the association of sociodemographic, behavioral, and physiological factors related to hypertension in patients seeking care at the different healthcare facilities in Kenya.

### Methodology

We carried out a cross-sectional survey study between April 2023 and July 2023. The general adult public visiting the outpatient clinics were recruited from 8 healthcare facilities in Kenya. Summary statistics were presented as medians and interquartile ranges for continuous data and frequencies and percentages for categorical data. The non-parametric Kruskal–Wallis test was used to compare the continuous variables and Fisher's exact test was used to compare the categorical variables between group associations.

### Results

A total of 1444 clients were recruited and included in the analysis. The median age of participants was 47.0 years, 54.3% were females, 75.1% were married and 42.9% reported living in the rural areas. The prevalence of hypertension was found in 29.4% of clients, of which 48.5% lacked awareness of their diagnosis. Of the patients who

**Data availability statement:** All relevant data are within the paper and its Supporting information files.

**Funding:** The author(s) received no specific funding for this work.

**Competing interests:** The authors have declared that no competing interests exist.

knew of their diagnosis (n = 412), 53.1% did not achieve blood pressure control as defined by Joint National Committee on prevention, detection, evaluation, and treatment of high blood pressure. Of note, 39.1% of participants with hypertension were from faith-based health facilities, 31.5% were from public institution and 29.4% were from private institutions (p < 0.001). In rural areas, faith-based facilities are the dominant care providers. Type of facility, age, gender, education, marital status, body mass index and residence were associated with hypertension (p < 0.05).

## Conclusion

The study highlights a significant burden of hypertension among adults attending outpatient clinics in Kenya. Nearly half of hypertensive individuals were unaware of their condition, and among those diagnosed, more than half did not achieve target blood pressure levels, indicating gaps in screening, awareness, and management. These findings emphasize the need for targeted interventions, including improved screening, awareness campaigns, and enhanced treatment strategies to improve hypertension control in diverse healthcare settings.

## Introduction

Hypertension is the most important risk factor for cardiovascular disease and remains the leading cause of mortality worldwide. According to the World Health Organization (WHO), an estimated 1.28 billion adults suffer from hypertension worldwide causing approximately 7.6 million premature deaths [1]. Globally, about half of all adults with hypertension are unaware of their condition and only about one in five individuals with hypertension have their blood pressure under control. The burden has shifted to low- and middle-income countries (LMICs), which now account for 66% of people with hypertension [1,2]. The Pan-African Society of Cardiology has identified hypertension as the top priority of action to help curb and reduce the incidence of heart disease and stroke on the African continent. However, very few countries in sub-Saharan Africa (SSA) have developed and implemented clear hypertension policies related to diagnosis and management of hypertension [3].

In SSA, hypertension, is a growing public health problem [3]. Studies have identified associated factors such as older age, higher body mass index (BMI), diabetes, lower and higher level of education, family history, and alcohol and tobacco use. Caffeine abuse has also been noted to have a negative effect on the prevalence of hypertension [4]. Okello et al, reported a 25.4% hypertension prevalence across seven SSA communities, with only half aware of their diagnosis, 50.5% of those diagnosed were on treatment, and just 47.3% of those treated achieving blood pressure control [5]. These findings suggest substantial room for improvement in hypertension diagnosis and treatment.

In Kenya, hypertension prevalence among adults aged 18–69 years is estimated at 24% − 29% whereas 50% are estimated to be pre-hypertensive [2,6–9]. Notably, only about a quarter of the affected patients are aware of their diagnosis, only about

18% receive treatment, and about 7% achieve hypertension (blood pressure) control [10]. Despite the strong evidence demonstrating that hypertension treatment reduces the incidence of CVD and lowers the mortality rates, this has not been effectively implemented in practice among SSA [10]. Pengpid and Peltzer reported a 28.6% prevalence, with 29.4% of individuals aware of their diagnosis, 6.5% currently using antihypertensive medication and only 12.5% having blood pressure control less than 140/90 mmHg [8]. Similarly, Mohammed et al. found a 24.5% (age standardized) prevalence, with 15.6% aware of their diagnosis, 27% receiving treatment and approximately 50% achieving blood pressure control [9]. Another study looking at the prevalence of hypertension in slum dwellers in Nairobi, also found that the prevalence of hypertension in this specific population was about 12% with about 20% being aware of their diagnosis, 47% were on some sort of treatment and 27% had blood pressures less than 140/90 mmHg [11]. In Kenya overall, one in ten women suffer from hypertension and only half of the populations with hypertension are aware of their diagnosis, indicating that there is a high burden of undiagnosed and uncontrolled high blood pressure in our population as SSA [12].

This study aims to explore the prevalence of hypertension, and the association of sociodemographic, behavioral, and physiological factors related to hypertension in clients seeking care at the different healthcare facilities (including public, private, and faith-based) in Kenya.

## Materials and methods

We carried out a cross-sectional survey study between 1st April 2023 and 31st July 2023. The study was conducted at eight healthcare facilities, grouped into private, public, and faith-based hospitals. The private hospitals were Aga Khan University Hospital, Nairobi; Aga Khan Hospital, Mombasa; and Avenue Hospital, Nairobi. The public hospitals were Garissa County Referral Hospital, Garissa; Kericho County Referral Hospital, Kericho; and Kenyatta University Teaching, Referral and Research Hospital, Nairobi. The faith-based hospitals were PCEA Chogoria Mission Hospital, Tharaka-Nithi, and Tenwek Hospital, Bomet. All the participating facilities provided either secondary or tertiary levels of care.

The study population was any general adult public visiting the outpatient clinics at the different healthcare facilities. The only eligibility criteria was adults ≥ 18 years of age and those who consented to be part of the study. Based on an assumption of 24.5% [8] and an attrition rate of 15%, the minimum sample size calculated was 328 participants. Since we were recruiting from three types of facilities (private, public and faith-based), the minimum sample required was 984 participants. The participants were recruited based on convenience sampling.

The survey was in English and consisted of questions based on demographics and clinical characteristics, history of hypertension and current antihypertensive medication use. To reduce potential bias of self-reported data, confidentiality of participants and privacy of their responses was prioritized. Data were collected both through online and paper-based surveys. Each site had an investigator that was responsible for the administration of the survey. The site investigator, based on the resources available at the respective sites, decided to choose either using paper or online surveys. Each site investigator received preliminary training on the administration of the survey. The site investigator interviewed the patient and was responsible for filling in the survey. Online survey data were collected through the Research Electronic Data Capture (REDCap) platform (Vanderbilt and National Institute of Health) [13]. Online and written consent was obtained from all the participants. Approval for this study was obtained from the Institutional Scientific and Ethics Review Committee (ISERC) (Ref:2022/ISERC-81(v3)) at the Aga Khan University, Nairobi, Kenya's National Commission for Science, Technology and Innovation (NACOSTI) (Ref:217822) and from the administrative leadership of all participating hospitals. Participants were allowed to withdraw from the study at any time without any consequences.

Sociodemographic factors included age, gender, race, education, marital status, income, and residence. Behavioral risk factors included body mass index (BMI), current daily smoker, current daily consumer of alcohol, adequate physical activity and insufficient fruit and vegetable intake. BMI was calculated as weight in kilograms divided by the square of height in meters (kg/m²). Adequate physical activity was defined as engaging in ≥ 3 days of vigorous activity of at least 20 minutes per day or ≥ 5 days of moderate intensity activity or walking of at least 30 minutes per day [14]. Insufficient fruit and

vegetable intake was defined as consuming <5 servings of fruits and/or vegetables per day [15]. Diabetes was reported as a prior diagnosis of diabetes by a health-care professional.

Blood pressures were measured, and three readings were taken on the left arm in a seated position, at 1-minute intervals. A large or extra-large cuff was primarily used based on the participant bicep diameter. All sites were encouraged to use either an OMRON basic BP machine or a Mindray BP machine based on the availability at the specific clinic. An average of the three readings was taken as the final value for both systolic and diastolic blood pressures. Hypertension at clinic visit was defined by JNC-7 as mean systolic blood pressure (SBP) at least 140 mmHg, or diastolic blood pressure (DBP) at least 90 mmHg [16]. Clients recruited included both individuals with and without a prior diagnosis of hypertension.

Continuous data were analyzed using medians and interquartile ranges (IQRs) whereas categorical data were analyzed as frequencies and percentages. The non-parametric Kruskal–Wallis test was used to compare the continuous variables and Fisher's exact test was used to compare the categorical variables between group associations. Logistic regression was also employed to identify associations and odds ratio, and 95% CI was presented. Data analysis was performed using SPSS statistical software V. 20.0 (IBM, Armonk, NY, USA). The significance level was set at $\alpha = 0.05$, and all tests were two-tailed.

## Results

### Demographic characteristics

A total of 1444 patients were recruited and included in the analysis. Table 1 presents the overall demographic characteristics of the participants. The median age of participants was 47.0 years (IQR: 34.0, 60.0) with 54.3% being females. More than half of the participants (75.1%) were married and about 70.6% had an education of secondary or higher. Most of the participants were Africans (96.4%), 65.4% were employed or self-employed and about 42.9% reported living in the rural areas and the recruitment covered residence from 33 counties.

### Behavioral characteristics and disease prevalence

Table 2 summarizes the behavioral characteristics. About half, 54.8% were overweight or obese. About 6.4% reported being current daily smokers whereas 8.1% reported being current daily alcohol users. Less than half, 44.4% reported having adequate physical activity and 48.7% reported having sufficient intake of fruits and vegetables. Based on comorbid conditions, 13.5% reported having diabetes, 12.8% reported having cardiovascular diseases and 28.5% reported having a prior diagnosis of hypertension. High systolic blood pressure was recorded in 29.4% of patients, whereas 17.3% had high diastolic pressure. Based on the average measures of systolic and diastolic blood pressures, the prevalence of hypertension was found in 29.4% of the clients visiting the healthcare facilities.

### Risk factors associated with hypertension

Table 3 summarizes the association between hypertension with demographic and behavioral characteristics, including odds ratios with corresponding 95% confidence intervals. Based on demographic characteristics, age, gender, education, marital status, and residence were associated with hypertension. The median age for those with hypertension was higher as compared to those with no hypertension (55.0 years vs 42.0 years; OR: 1.04 (95% CI: 1.04–1.05); p<0.001). More males were reported to have hypertension as compared to not having hypertension (52.4% vs 42.9%; OR: 1.46 (95% CI: 1.16–1.83); p=0.001). More patients who had hypertension reported having lower education (primary or less) as compared to those who did not have hypertension (37.3% vs 26.1%; p<0.001). More patients that had hypertension reported living in the rural areas as compared to those not having hypertension (51.5% vs 39.3%; OR: 1.64 (95% CI: 1.31–2.06); p<0.001). The type of healthcare facilities were associated with hypertension, where 39.1% of participants

**Table 1. Overall demographic characteristics of 1444 patients.**

| Socio-Demographic Factors | | n | % |
|---|---|---|---|
| Study Site | Private Facility | 557 | 38.6% |
| | Public Facility | 160 | 31.2% |
| | Faith Based Facility | 195 | 30.3% |
| Age (years) (n = 1427) (median [IQR]) | | 47.0 [34.0, 60.0] | |
| Sex (n = 1442) | Female | 783 | 54.3% |
| | Male | 659 | 45.7% |
| Race (n = 1441) | African | 1389 | 96.4% |
| | Asian | 30 | 2.1% |
| | Caucasian | 13 | 0.9% |
| | Others (Arab/Somali) | 9 | 0.6% |
| Education (n = 1423) | No School | 180 | 12.6% |
| | Primary | 239 | 16.8% |
| | Secondary | 377 | 26.5% |
| | Tertiary | 627 | 44.1% |
| Marital Status (n = 1440) | Married | 1081 | 75.1% |
| | Others | 92 | 6.4% |
| | Single | 267 | 18.5% |
| Employment (n = 1438) | Employed/ Self Employed | 941 | 65.4% |
| | Retired | 175 | 12.2% |
| | Unemployed | 322 | 22.4% |
| Income Per Month (Kenya Shillings) (Gross Amount) | 0 - 15,000 | 395 | 27.4% |
| | 100,000 - 200,000 | 115 | 8.0% |
| | 16,000 - 50,000 | 235 | 16.3% |
| | 51,000 - 100,000 | 202 | 14.0% |
| | Above 200,000 | 44 | 3.0% |
| | Prefer not to answer | 453 | 31.4% |
| Place of Residence (n = 1435) | Rural | 616 | 42.9% |
| | Urban | 819 | 57.1% |
| Regions (n = 1428) | Central | 202 | 14.1% |
| | Coast | 160 | 11.2% |
| | Eastern | 232 | 16.2% |
| | Nairobi | 269 | 18.8% |
| | North Eastern | 182 | 12.8% |
| | Nyanza | 33 | 2.3% |
| | Rift Valley | 350 | 24.5% |

with hypertension were from faith-based institutions, 31.5% from public institution and 29.4% from private institutions (p < 0.001).

Based on behavioral characteristics, BMI, alcohol consumption, food intake, diabetes, cardiovascular disease, and prior diagnosis of hypertension were associated with hypertension. Among participants with hypertension, 65.4% were either overweight or obese, compared to 50.4% of those without hypertension (p < 0.001). Daily alcohol consumption was more common among hypertensive participants than non-hypertensive participants (10.6% vs. 7.1%; OR: 1.56 (95% CI: 1.05–2.30); p = 0.026). Sufficient fruit and vegetable intake was reported by 39.5% of hypertensive participants, compared to 52.5% of those without hypertension (p < 0.001). Hypertension was also associated with comorbidities such as diabetes

**Table 2. Overall behavioral characteristics of the study cohort.**

| Behavior Risk Factors | | n | % |
|---|---|---|---|
| Body Mass Index (BMI) (n = 1413 | Underweight | 83 | 5.9% |
| | Healthy | 555 | 39.3% |
| | Overweight | 457 | 32.3% |
| | Obese | 318 | 22.5% |
| Current Daily Smoker | No | 1351 | 93.6% |
| | Yes | 93 | 6.4% |
| Current Daily Consumer of Alcohol | No | 1327 | 91.9% |
| | Yes | 117 | 8.1% |
| Adequate Physical Activity | No | 803 | 55.6% |
| | Yes | 641 | 44.4% |
| Sufficient Fruits and Vegetable Intake | No | 741 | 51.3% |
| | Yes | 703 | 48.7% |
| Diabetes | No | 988 | 68.4% |
| | Yes | 195 | 13.5% |
| | Unknown | 261 | 18.1% |
| Cardiovascular Disease | No | 936 | 64.8% |
| | Yes | 185 | 12.8% |
| | Unknown | 323 | 22.4% |
| Prior Diagnosis of Hypertension | No | 1032 | 71.5% |
| | Yes | 412 | 28.5% |
| Current use of prescribed medication | No | 59 | 14.3% |
| | Yes | 353 | 85.7% |
| Overall Hypertension | No | 1019 | 70.6% |
| | Yes | 425 | 29.4% |

and cardiovascular disease (p < 0.001 for both). Of the 425 with elevated blood pressure at the clinic visit, 51.5% reported a prior diagnosis of hypertension, compared to 18.9% among those without elevated blood pressure.

## Discussion

This study found a high prevalence of hypertension (29.4%) among the study population, with almost half (48.5%) of hypertensive participants receiving a first-time diagnosis during the survey. Hypertension was more prevalent among rural residents compared to urban residents, and married individuals had a higher prevalence than unmarried participants. Older age, male gender, and lower education levels were associated with higher hypertension prevalence. Lifestyle factors such as low fruit and vegetable intake, physical inactivity, higher alcohol consumption, and higher rates of overweight/obesity were more common among hypertensive individuals.

### High prevalence of hypertension with low awareness

Africa has the highest global hypertension prevalence, with an estimated 150 million adults affected – a number projected to reach 216.8 million by 2030 [17–20]. In Kenya, CVD mortality (13.8%) is higher than the global average (12.8%) [20,21]. Our study found a prevalence of 29.4%, consistent with other Kenyan studies (24.5–30%) [7,8,22]. Non-communicable diseases (NCDs), including CVD, already account for 50% of hospitalisations and 55% of inpatient deaths in Kenya [23], making prevention and early detection urgent priorities.

**Table 3. Association between hypertension with demographic and behavioral characteristics.**

| | | Hypertension | | | | OR (95% CI) | P Value |
|---|---|---|---|---|---|---|---|
| | | NO | | YES | | | |
| | | (n = 1019) | | (n = 425) | | | |
| Age (years) (n = 1427) (median [IQR]) | | 42.0 [32.0, 57.0] | | 55.0 [43.0, 65.0] | | 1.04 (1.04–1.05) | <0.001 |
| Sex (n = 1442) | Female | 581 | 57.1% | 202 | 47.6% | Reference | |
| | Male | 437 | 42.9% | 222 | 52.4% | 1.16 (1.16–1.83) | 0.001 |
| Race (n = 1441) | African | 977 | 96.1% | 412 | 97.2% | 1.41 (0.73–2.71) | 0.308 |
| | Others | 40 | 3.8% | 12 | 2.8% | Reference | |
| Education (n = 1423) | No School | 123 | 12.2% | 57 | 13.6% | 1.74 (1.20–2.51) | 0.003 |
| | Primary | 140 | 13.9% | 99 | 23.7% | 2.65 (1.92–3.66) | <0.001 |
| | Secondary | 247 | 24.6% | 130 | 31.1% | 1.97 (1.48–2.63) | <0.001 |
| | Tertiary | 495 | 49.3% | 132 | 31.6% | Reference | |
| Marital Status (n = 1440) | Married | 731 | 71.9% | 350 | 82.5% | 2.98 (2.06–4.31) | <0.001 |
| | Others | 55 | 5.4% | 37 | 8.7% | 4.18 (2.43–7.19) | <0.001 |
| | Single | 230 | 22.6% | 37 | 8.7% | Reference | |
| Employment (n = 1438) | Employed/ Self Employed | 673 | 66.2% | 268 | 63.5% | Reference | |
| | Retired | 110 | 10.8% | 65 | 15.4% | 1.48 (1.06–2.08) | 0.022 |
| | Unemployed | 233 | 22.9% | 89 | 21.1% | 0.86 (0.72–1.27) | 0.772 |
| Place of Residence (n = 1435) | Rural | 397 | 39.3% | 219 | 51.5% | 1.64 (1.31 −2.06) | <0.001 |
| | Urban | 613 | 60.7% | 206 | 48.5% | Reference | |
| Current Daily Smoker | No | 961 | 94.3% | 390 | 91.8% | Reference | |
| | Yes | 58 | 5.7% | 35 | 8.2% | 1.48 (0.96–2.30) | 0.074 |
| Current Daily Consumer of Alcohol | No | 947 | 92.9% | 380 | 89.4% | Reference | |
| | Yes | 72 | 7.1% | 45 | 10.6% | 1.56 (1.05–2.30) | 0.026 |
| Adequate Physical Activity | No | 552 | 54.2% | 251 | 59.1% | Reference | |
| | Yes | 467 | 45.8% | 174 | 40.9% | 0.82 (0.65–1.03) | 0.089 |
| Sufficient Fruits and Vegetable Intake | No | 484 | 47.5% | 257 | 60.5% | Reference | |
| | Yes | 535 | 52.5% | 168 | 39.5% | 0.59 (0.47–0.74) | <0.001 |
| Diabetes | No | 718 | 70.5% | 270 | 63.5% | Reference | |
| | Yes | 181 | 17.8% | 80 | 18.8% | 1.66 (1.21–2.29) | 0.002 |
| | Unknown | 120 | 11.8% | 75 | 17.6% | 1.18 (0.87–1.59) | 0.288 |
| Cardiovascular Disease | No | 713 | 70.0% | 223 | 52.5% | Reference | |
| | Yes | 220 | 21.6% | 103 | 24.2% | 3.68 (2.66–5.10) | <0.001 |
| | Unknown | 86 | 8.4% | 99 | 23.3% | 1.50 (1.13–1.98) | 0.004 |
| Prior Diagnosis of Hypertension | No | 826 | 81.1% | 206 | 48.5% | Reference | |
| | Yes | 193 | 18.9% | 219 | 51.5% | 4.55 (3.56–5.82) | <0.001 |
| BMI (n = 1413) | Underweight | 73 | 7.3% | 10 | 2.4% | 0.43(0.21–0.85) | 0.015 |
| | Healthy | 420 | 42.3% | 135 | 32.2% | Reference | |
| | Overweight | 320 | 32.2% | 137 | 32.7% | 1.33 (1.01–1.76) | 0.044 |
| | Obese | 181 | 18.2% | 137 | 32.7% | 2.36 (1.75–3.16) | <0.001 |
| Study Site | Private | 432 | 42.4% | 125 | 29.4% | 0.68 (0.51–0.91) | 0.008 |
| | Public | 316 | 31.0% | 134 | 31.5% | Reference | |
| | Faith | 271 | 26.6% | 166 | 39.1% | 1.45 (1.09–1.91) | 0.010 |

Almost half (48.5%; n = 206/425) of the hypertensive patients had a first-time diagnosis, which was much higher than findings from similar studies in Kenya 15.6% to 29.6% [8,9]. This highlights persistent low awareness, consistent with evidence from LMICs, where only 1 in 3 people are aware of their hypertension status [24]. The lack of awareness found in this study could be explained by a lack of screening, especially in the poorer populations of Kenya. Lack of screening—particularly in poorer, rural areas—remains a key barrier, worsened by transport costs, absence of National Health Insurance Fund coverage for testing, and limited health facility access [23]. It is important to increase hypertension screening in rural facilities, including governmental and Faith Based Hospitals (FBH's) as these facilities provide relatively inexpensive services. These FBH's provide 30% to 70% of healthcare services in LMIC's [25] and up to 40% in rural areas of Kenya [26]. The importance of such facilities is highlighted by the fact that the majority of hypertensive patients in this study (39.1%) were enrolled from a faith-based hospital. While these hospitals are not the only health facilities in their respective counties, their role as major care providers in rural areas may explain the higher representation observed in our sample.

## Urban – rural differences

Although most participants were urban residents, over 40% of hypertensive patients lived in rural areas, echoing the narrowing urban–rural gap in hypertension prevalence [8,27], as well as the results of a recent meta-analysis showing higher increases in prevalence in the rural populations of the most developed LMIC's [28]. One budding hypothesis for the higher prevalence in rural areas may be increased indoor air pollution from burning solid fuels in rural households. The incomplete combustion of fuels used for cooking and heating can lead to increased indoor particulate matter (PM) pollution which is positively correlated to the development of hypertension [29,30]. Many rural homes in Kenya still rely on the burning of solid fuels such as wood [31], compared to cleaner or more energy-efficient fuels such as natural gas or electricity used in urban homes [32]. This underlines a need to increase awareness on the health effects of indoor air pollution from solid fuels especially in rural Kenya.

## Demographic factors and hypertension

Our study identified several demographic factors—marital status, age, gender, and education—associated with hypertension. A large proportion of married participants (82.5%) were hypertensive, similar to findings from Nigeria where married men and women had an 88% increased likelihood of hypertension [33]. In contrast, some studies suggest marriage is associated with better health including lower cardiovascular risk factors [34] and with lower night-time systolic blood pressure, a finding which was more significant in men [35]. The relationship between marital status and hypertension prevalence seemingly varies by gender and has been reported by other studies. Aziz et al. reported married men were less likely to have higher blood pressures [36], while another study in Iran found men who were never married had up to a 55% increased risk for hypertension as compared to married men [37]. Meanwhile, Tuoyire et al. found that women who were married, cohabiting or previously married, had significantly higher chances of developing hypertension, concluding that marital status was an independent risk factor for hypertension for women in Ghana [38]. Thus, these findings may suggest marriage has more beneficial effect on men's health than women, and potentially even a disadvantageous effect on women's health. Looking at our own results, however, there is a higher likelihood of hypertension in both men and women who are married, with a larger percentage (49.7%) of hypertensive married participants being male. Cultural and gender roles may partly explain these differences—men may face greater pressure to provide financially, while women may juggle caregiving responsibilities alongside employment. Alternatively, marriage may increase socioeconomic status, enabling greater access to unhealthy processed foods linked to hypertension [33,38].

Age was also significantly associated with hypertension, with hypertensive participants having a higher median age (55 years) than non-hypertensive participants (42 years), consistent with other studies [8,9,27]. Since hypertension risk increases with age, the growing aging population in Africa and Kenya suggest a rising future burden of hypertension and related CVD's [39,40]. That said, hypertension has also been recorded in younger age groups (18–35 years) [41], and

33% of primary school children in urban Kenya have been observed to have stage 1 hypertension (prehypertension) [42]. Healthcare facilities and future research should therefore not favour age as a main factor for the screening of hypertension.

Gender differences were noted and this study found a higher percentage of men with hypertension than women. This is a common finding [8,18], some evidence even suggesting men twice more likely to have hypertension than women in Kenya [22]. It is suggested that women may get to know about their hypertension status earlier than men, possibly through antenatal screening or have better health seeking than men [8,24]. There is also a proposed protective role of oestrogen in the pathophysiology of hypertension, that reduces after menopause [43].

Prevalence of hypertension increases with increasing education level, however, only up to secondary education, after which the trend reverses. The highest level of education in this study, tertiary, was associated with the lowest prevalence of hypertension. Nearly half (49.3%) of non-hypertensive participants had tertiary education, whereas 68.4% of hypertensive participants had secondary education or lower. Evidence is mixed, with some studies reporting higher hypertension prevalence among those with higher education [8,44,45], while others show higher educational attainment linked to better health outcomes [46] including lower hypertension prevalence [47,48]. In Kenya, households headed by more educated individuals were more likely to utilise healthcare services, possibly due to greater awareness of the importance of regular health checkups [49]. Additionally, conditions such as hypertension and its risk factors may be relatively unknown in people with little education [50] possibly leading to less knowledge of healthier lifestyles for prevention of hypertension, or the importance of early screening and intervention.

## Behavioral factors and hypertension

In both the hypertensive and non-hypertensive groups, more than half reported inadequate daily physical activity. Physical inactivity increases the risk of developing hypertension [51] and exercise can not only prevent or delay the onset of hypertension [52], but the reduction in blood pressure from increased physical activity can be as great as first-line antihypertensive medications [53]. The concerning lack of physical activity in this cohort extends to other studies done in Kenya, including one finding 80.3% of the population did not have sufficient physical activity [54], and another finding 86.8% of adolescents in the country are insufficiently active [20]. These results highlight the need for public education on the health implications of inactivity and for greater emphasis on physical education in schools.

Dietary differences were also evident between the groups. A lower percentage of the patients with hypertension had sufficient fruit and vegetable intake compared to non-hypertensive patients. Increased fruit and vegetable consumption promotes a healthy diet and possibly leads to less consumption of unhealthy foods high in fat and salt. An inverse relationship between the consumption of fruit and vegetable and hypertension risk has been shown [55]. This is postulated to be because fruit and vegetable are high in fibre and essential vitamins and minerals such as potassium, magnesium, vitamin C and others [56,57]. Increased dietary fibre has been shown to reduce blood pressure [57], and higher potassium intake is linked to a lower prevalence of hypertension [58]. Alcohol intake was another important factor -moderate to heavy drinking increases hypertension risk [59], and not drinking alcohol was seen to lower the risk of hypertension in 18–35-year-old Kenyans by 70% [41]. In this study, a higher percentage of hypertensive patients were daily consumers of alcohol compared to non-hypertensive patients.

Supporting what our study showed, physical inactivity and poor diet are major contributors to overweight/obesity which has been known for a long time to be an independent risk factor for hypertension [1,60]. In this study there was a higher percentage of obese patients in the hypertensive group. Higher BMI's are associated with hypertension [8] and in LMIC's the fastest increase in obesity comes from rural rather than urban areas [28]. The likelihood of an overweight/obese person having hypertension is 8 times compared to a healthy weight individual [27]. Among primary school children in Kenya, overweight status was associated with a 1.83-fold increased risk for hypertension compared to healthy weight peers [42]. The prevalence of obesity in Kenya was found at 31.13% [61], however with continuing urbanisation and shift from traditional to more westernised diets, this prevalence is likely to increase [62].

This study had several limitations. Firstly, the blood pressures were measured in one clinic setting and ideally would have been better to measure them in various or multiple clinic visits. Secondly, even though we have 8 participating sites, additional sites especially county hospitals, where most Kenyans seek medical attention, would have been more reflective of the burden of hypertension in Kenya.

### Future considerations

It is important to consider factors associated with hypertension prevalence in more detail in the Kenyan population. For example: while fruit and vegetable intake is low, high sodium consumption is also a key dietary driver of hypertension [57]. Although this study measured daily consumption of alcohol, future research should assess the number of units consumed to better evaluate its impact.

The PASCAR and WHF cardiovascular scorecard project found that 72.5% of people in Kenya live in rural areas [20]. However, rural residence does not necessarily equate to low income, and future studies should explore how income levels in both rural and urban settings relate to hypertension prevalence. Beyond conventional lifestyle factors such as poor diet and inactivity, psychosocial stress is known to increase both hypertension prevalence and poor control [63]; assessing stress alongside behavioural and sociodemographic characteristics could provide a more complete understanding of risk.

Education also warrants closer investigation. Our findings suggest a reversed trend in hypertension prevalence between secondary and tertiary education, with some evidence indicating a stronger link between education and hypertension in women [48]. Further gender-based comparisons, particularly among married individuals, could clarify these associations.

In addition, exploration of prehypertension and its progression into hypertension in the Kenyan population may inform more accurate intervention measures. Previous studies estimate the prevalence of prehypertension in Kenya at 54.5% [64] and it has even been documented among primary school children [42] Given its association with increased risk for CVD's [65], screening for prehypertension in clinical settings should be encouraged to allow for earlier health interventions to prevent conversion to hypertension.

### Conclusion

This study found a high prevalence of hypertension in the population, especially in rural areas of Kenya, with low awareness. Factors associated with hypertension were higher age group, being male, being married, having secondary or lower education, insufficient consumption of fruit and vegetable, obesity and alcohol consumption. Primary hypertension is preventable and blood pressure tests are simple, quick, non-invasive and should be inexpensive. From this study and others done in Kenya, it is clear that public education strategies should be implemented to raise awareness on hypertension and its risk factors, with special focus on rural areas. Along with this, public health interventions to promote healthier lifestyles to all age groups in Kenya are vital in trying to control the rising burden of hypertension and related CVDs. It is also crucial to strengthen hypertension screening in the country, not only within health facilities – where blood pressure assessment is already part of routine outpatient services—but also at the community level through initiatives led by the Ministry of Health in collaboration with county health departments. Community-based strategies, such as integration of blood pressure checks into existing outreach programs, screening during community health campaigns, and use of community health volunteers, could expand early detection and intervention. The fact remains that hypertension disease burden in Kenya will continue rising with increasing urbanisation and development of the country, its control must therefore be made a national priority.

### Supporting information

**S1 Table. Anonymous data supporting the results.**

(XLS)

## Author contributions

**Conceptualization:** Jasmit Shah, Caroline Mithi, Sayed K Ali.

**Data curation:** Jasmit Shah, Soraiya Manji, Cynthia Smith.

**Formal analysis:** Jasmit Shah.

**Investigation:** Jasmit Shah, Jamila Nambafu, Anthony Ochola, Linda Barasa, Faraj Amir, Husni Abdalla, Simeon Jowi, Caroline Mithi, Rajiv Patel, Sayed K Ali.

**Methodology:** Jasmit Shah, Soraiya Manji, Cynthia Smith, Jamila Nambafu, Anthony Ochola, Linda Barasa, Faraj Amir, Husni Abdalla, Simeon Jowi, Caroline Mithi, Rajiv Patel, Sayed K Ali.

**Project administration:** Soraiya Manji, Jamila Nambafu.

**Supervision:** Jasmit Shah, Soraiya Manji, Jamila Nambafu, Anthony Ochola, Linda Barasa, Faraj Amir, Husni Abdalla, Simeon Jowi, Caroline Mithi, Rajiv Patel, Sayed K Ali.

**Validation:** Jasmit Shah, Cynthia Smith, Anthony Ochola, Linda Barasa, Faraj Amir, Husni Abdalla, Simeon Jowi, Caroline Mithi, Rajiv Patel, Sayed K Ali.

**Writing – original draft:** Jasmit Shah.

**Writing – review & editing:** Jasmit Shah, Soraiya Manji, Cynthia Smith, Jamila Nambafu, Anthony Ochola, Linda Barasa, Faraj Amir, Husni Abdalla, Simeon Jowi, Caroline Mithi, Rajiv Patel, Sayed K Ali.

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
