## [Decision Letter · Decision Letter 0]

7 Aug 2025

Dear Dr. Shah,

Thank you for submitting your manuscript to PLOS ONE. After careful consideration, we feel that it has merit but does not fully meet PLOS ONE’s publication criteria as it currently stands. Therefore, we invite you to submit a revised version of the manuscript that addresses the points raised during the review process.

We look forward to receiving your revised manuscript.

Kind regards,

Sandra Boatemaa Kushitor, Ph.D.

Academic Editor

PLOS ONE

Journal Requirements:

3. We note that there is identifying data in the Supporting Information file <Hypertension Data.xls>. Due to the inclusion of these potentially identifying data, we have removed this file from your file inventory. Prior to sharing human research participant data, authors should consult with an ethics committee to ensure data are shared in accordance with participant consent and all applicable local laws.

-Location data

Please remove or anonymize all personal information, ensure that the data shared are in accordance with participant consent, and re-upload a fully anonymized data set. Please note that spreadsheet columns with personal information must be removed and not hidden as all hidden columns will appear in the published file.

**Additional Editor Comments:**

Dear Authors,

Kindly revise the manuscript based on the comments provided.

Reviewers' comments:

Reviewer's Responses to Questions

**Comments to the Author**

1. Is the manuscript technically sound, and do the data support the conclusions?

Reviewer #1: Yes

2. Has the statistical analysis been performed appropriately and rigorously?

Reviewer #1: Yes

3. Have the authors made all data underlying the findings in their manuscript fully available?

Reviewer #1: Yes

4. Is the manuscript presented in an intelligible fashion and written in standard English?

Reviewer #1: Yes

Reviewer #1: Overall, the manuscript is well-written. The aim of the study is clearly described. The methods are aligned with the study aim. The results are reported and discussed satisfactorily.

Introduction

• Paragraphs 2 and 3 are a repetition of the same point. Could be combined in one paragraph or summarised.

Methods

• Clearly described. Methods are aligned with the study aim.

Results

• The planned sample size was 328 but 1444 participants were recruited. What was the rationale for increasing the sample size 4 times.

• Significant findings (hypertension was associated with obesity and age) were not stated in the Results section. Please provide narrative describing the data in the tables.

• The abbreviation FV is not widely used. Consider writing in full.

Conclusion

• The conclusions answer the research question.

References

• Please format the references according to the journal guidelines and include DOIs.

**Do you want your identity to be public for this peer review?** For information about this choice, including consent withdrawal, please see our Privacy Policy

Reviewer #1: **Yes: ** Keshena Naidoo

---

## [Author Response · Author response to Decision Letter 1]

12 Aug 2025

Response to Comments

Overall Editor Comments

Journal Requirements:

Response: We have revised the manuscript to the journals style requirements. We have incorporated from both the templates.

Response: We have included the captions.

3. We note that there is identifying data in the Supporting Information file <Hypertension Data.xls>. Due to the inclusion of these potentially identifying data, we have removed this file from your file inventory. Prior to sharing human research participant data, authors should consult with an ethics committee to ensure data are shared in accordance with participant consent and all applicable local laws. Data sharing should never compromise participant privacy. It is therefore not appropriate to publicly share personally identifiable data on human research participants. Please remove or anonymize all personal information, ensure that the data shared are in accordance with participant consent, and re-upload a fully anonymized data set. Please note that spreadsheet columns with personal information must be removed and not hidden as all hidden columns will appear in the published file.

Response: We have revised and ensured the data file is completely anonymous.

Response: We appreciate the reviewer’s comments as they have helped us to strengthen our manuscript.

Reviewer #1:

Introduction

1. Paragraphs 2 and 3 are a repetition of the same point. Could be combined in one paragraph or summarised.

Response: We have revised the introduction for better flow.

Methods

2. Clearly described. Methods are aligned with the study aim.

Response: Thank you for the comment.

Results

3. The planned sample size was 328 but 1444 participants were recruited. What was the rationale for increasing the sample size 4 times.

Response: Based on an assumption of 24.5% and an attrition rate of 15%, the minimum sample size calculated was 328 participants. Since we were recruiting from three types of facilities (private, public and faith-based), the minimum sample required was 984 participants. We have revised this in the methods section.

4. Significant findings (hypertension was associated with obesity and age) were not stated in the Results section. Please provide narrative describing the data in the tables.

Response: This has been revised in the results section

5. The abbreviation FV is not widely used. Consider writing in full.

Response: We have revised and written it in full.

Conclusion

6. The conclusions answer the research question.

Response: Thank you for the comment.

References

7. Please format the references according to the journal guidelines and include DOIs.

Response: We have revised the references accordingly.

If reviewer comments were submitted as an attachment file, they will be attached to this email and accessible via the submission site

Response: Furthermore, the reviewer had suggested few changes and comments to the manuscript. Below are the comments and our responses.

1. The title is misleading. The sample are not people living with HPT, but clients attending the OPD. Change to “Prevalence and risk factors of hypertension among clients seeking care at selected facilities in Kenya”

Response: We have revised the Title.

Abstract Comments:

2. Abstract: Include a section about how the data was analysed.

Response: We have included a summary of the analysis in the Methods of the abstract.

3. Is goal blood pressure determined by the clients or medical doctors? Are you referring to hypertension control as directed by the WHO.

Response: We have included in the abstract the use of guidelines defined by the Joint National Committee.

4. This statement is misleading. The facility is faith-based not all the clients are pastors. Kindly rewrite for clarity.

Response: We have clarified the sentence.

Introduction Comments:

5. The switch from global, to regional and national level evidence from the literature is inconsistent. You have Kenya in the previous sections, Nigeria in this section and from line 89 you have global evidence. Use the funnel

Response: We have reworked the entire Introduction to flow it better.

6. I disagree with this assertion; there are countless studies on hypertension in SSA. Look at the RODAM project and EDULINK project

Response: We have omitted the sentence.

Methods Comments:

7. Long confusing sentence. Group them by location first; for example Nairobi. And only indicate the number that are private, public or faith-based.

Response: We have revised and grouped them based on the type of healthcare facility.

8. Include references for the behavioral risk factors.

Response: We have included the references.

9. Kindly describe how BMI was measured in the methods. BMI is a lifestyle or behavioural risk but not demographic characteristics

Response: We have described in the methods and put BMI with the other behavioral characteristics.

10. Start sentence with words.

Response: This has been revised.

11. Restructure the Results for Table 3.

Response: We have revised the results.

12. This statement is incorrect. Past history is hyp diagnosis.

Response: We have revised the sentences for better clarity.

Discussion

13. Kindly present your logistic regression before the discussion. Although you have presented the discussion in sections; it is best to present a key section where you summarise your results before providing the interpretation

Response: We have not done logistic regression for these analyses. We have excluded it from the methods. This was an oversight. We have tried to rework on the discussion for better structure.

References:

14. Most of the references are incomplete. Add the url link for the reports. For example, how can we access ref 4.

Responses: We have tried to incorporate and correct the references.

15. What is the heading for this table

Response: We have included the headings.

16. Why do the sample size vary across variables. This is very confusing. Keep only the same sample size throughout the study.

Response: Since participation in the survey was voluntary, some questions were left unanswered, resulting in missing responses for certain variables. To ensure transparency, we have reported the sample size (n) for each variable, allowing readers to see where data are missing.

17. This difference between 52 and 49 is quite small. This association will not be retained in a regression model. Kindly provide evidence of this in a regression model

Response: The observed difference is not between 51.5% and 48.5%, but rather between 51.5% of participants with elevated blood pressure who were from rural areas and 39.3% of participants without elevated blood pressure who were from rural areas. This represents a 12.2 percentage point difference.

---

## [Editor Report · Decision Letter 1]

12 Sep 2025

Dear Dr. Shah,

Thank you for submitting your manuscript to PLOS ONE. After careful consideration, we feel that it has merit but does not fully meet PLOS ONE’s publication criteria as it currently stands. Therefore, we invite you to submit a revised version of the manuscript that addresses the points raised during the review process.

We look forward to receiving your revised manuscript.

Kind regards,

Sandra Boatemaa Kushitor, Ph.D.

Academic Editor

PLOS ONE

Journal Requirements:

Additional Editor Comments:

Dear Authors,

Kindly revise the manuscript based on the comments in the attached file.

In the response to reviewer's file do only indicate "this correction has been made", copy and paste the revision. Your document should be such that decisions can be made based on it.

Kind regards,

---

## [Author Response · Author response to Decision Letter 2]

15 Sep 2025

We have revised all the edits and suggestions given by the reviewer as a track changed document. Below are our responses to specific comments and suggestions made specifically on the document.

Kindly revise the manuscript based on the comments in the attached file.

In the response to reviewer's file do only indicate "this correction has been made", copy and paste the revision. Your document should be such that decisions can be made based on it.

1. Revision from patients to clients

Response. This correction has been made.

2. Kindly consider how this is phrased. In rural areas, faith-based facilities are the dominant care providers. If you flip this around, htn is increasing in rural areas, which is what your findings indicate. This was not the case in the past and it would be great if you could highlight that

Response: We have rephrased and added “ In rural areas, faith-based facilities are the dominant care providers”

3. Hospital?

Response: The statement is correct. For each hospital, we have provided the location (city or town) where it is situated, for example, Aga Khan University Hospital, Nairobi.

4. Did you consider those with previous diagnosis?

Response: We considered both with and without previous diagnosis. We have revised in the statement where it was highlighted.

5. Kindly present the logistic regression results.

Response: We have added the Odds Ratio (OR) in Table 3 and revised the results section accordingly.

6. The data does not support this conclusion: only 72 participants were diagnosed by the study who had no previous diagnosis

Response: From the 1444 clients, 425 (29.4%) met the criteria for hypertension at clinic visit. Among the 425 clients that met the criteria for hypertension, 219 (51.5%) had a prior diagnosis of hypertension whereas 206 (48.5%) did not have a prior diagnosis. We have included the numbers in the paragraph, and this reflects in the table.

7. Another motivation could also be that these facilities provide relatively inexpensive services

Response: The correction has been made and have added this in the discussion.

8. Kindly indicate whether these are not the only hospitals in the counties. These centres may have their own pull factors, such as short waiting times compared to crowded public hospitals. Going to seek care at any facility is not a risk factor of hypertension

Response: We agree that the site of care is not a risk factor for hypertension. Our intention was to highlight the role of faith-based hospitals as major providers of care in rural Kenya, which may explain why the largest proportion of participants were recruited from these facilities. We have revised the text to clarify that these hospitals are not the only health facilities in their counties.

9. This was not part of your results

Response: We agree that the statement about prehypertension in primary school children was not part of our study results but was included as supporting evidence from previous literature. To avoid confusion, we have revised the text to clearly separate our findings from published data as this is presented in future considerations.

10. Which government unit should be doing this? Is blood pressure assessment part of regular OPD services? If yes, that’s facility level screening. Now find out how community level screening can be done and suggest that

Response: We have corrected and revised the paragraph.

---

## [Editor Report · Decision Letter 2]

24 Sep 2025

Prevalence and Risk Factors of Hypertension Among Clients Seeking Care at Selected Healthcare Facilities in Kenya

PONE-D-25-09467R2

Dear Dr. Shah,

We’re pleased to inform you that your manuscript has been judged scientifically suitable for publication and will be formally accepted for publication once it meets all outstanding technical requirements.

Kind regards,

Sandra Boatemaa Kushitor, Ph.D.

Academic Editor

PLOS ONE
---

## [Editor Report · Acceptance letter]

PONE-D-25-09467R2

PLOS ONE

Dear Dr. Shah,

I'm pleased to inform you that your manuscript has been deemed suitable for publication in PLOS ONE. Congratulations! Your manuscript is now being handed over to our production team.

Kind regards,

on behalf of

Dr. Sandra Boatemaa Kushitor

Academic Editor

PLOS ONE